# Study on Physical and Mechanical Properties of High-Water Material Made by Seawater

Bangwen Lu [1,2], Changwu Liu [3,4,*], Jungang Guo [1,2] and Naiqi Feng [1,2]

[1] Zhengzhou Institute of Multipurpose Utilization of Mineral Resources, Chinese Academy of Geological Sciences (CAGS), Zhengzhou 450006, China
[2] China National Engineering Research Center for Utilization of Industrial Minerals, Zhengzhou 450006, China
[3] College of Water Resource and Hydropower, Sichuan University, Chengdu 610065, China
[4] State Key Laboratory of Hydraulics and Mountain River Engineering, Chengdu 610065, China
[*] Correspondence: liuchangwu@scu.edu.cn; Tel.: +86-0371-68632048

**Abstract:** In maritime engineering, marine-derived construction materials are seen as an efficient and cost-effective alternative. HWM is a novel inorganic cementitious material characterized by its high water content, rapid setting, and early strengthening. In this study, first, HWM was proposed to be produced from seawater and used in a maritime environment. Two groups of HWM samples with varied w/c ratios were prepared with fresh water and seawater, and their behavior was examined to assess the viability of HWM produced with seawater. The microstructures and chemical compositions were studied using SEM and XRD. Results indicated that as the w/c ratio increased from 3:1 to 6:1, the water content, density, and uniaxial compressive strength of HWM produced from seawater varied from 72.1% to 77.5%; 1.25 to 1.12 $g/cm^3$, and 1.47 MPa to 0.39 MPa, respectively, which is 2–10% lower, 0.8–2.2% higher, and 13–45% stronger than that from fresh water. The chemical composition of HWM mixed with seawater is predominantly composed of ettringite, C-S-H gel, aluminum ($Al(OH)_3$) glue, M-S-H gel, and $Mg(OH)_2$. $SO_4^{2-}$ and $Mg^{2+}$ in seawater participate in the hydration and hardening of HWM, resulting in an increase in the synthesis of ettringite and M-S-H gel, which makes the skeletal structure of HWM denser, hence increasing its strength. HWM derived from seawater retains excellent physical and mechanical properties. This work reveals the HWM-seawater interaction mechanism, elucidates the promising application prospect of HWM in maritime engineering, and paves the way to investigate its field performance.

**Keywords:** high-water material; seawater; water-cement ratio; microstructure; water content; strength

## 1. Introduction

In recent years, with the expansion of ocean exploitation, the construction of maritime infrastructure has grown at an unparalleled rate [1]. Cementing material has been widely used and chosen among manmade construction materials in maritime environments around the world due to its low cost and ease of construction [2]. Cementing material draws a matter of continuing concern due to a high environmental cost. It is reported that cement production is the third-largest producer of $CO_2$ in the world after transport and energy generation [3]. Sustainability for cementitious material has attracted widespread interest and has become a hot topic of research [4–6]. Recently, recycled cementing materials in cementing materials production have become more and more popular in terms of less consumption of natural materials and many environmental advantages of disposal and reusing of waste materials. The behavior of recycled cementing materials was investigated by a large number of researchers [7–13]. However, the long-term durability of recycled cementing materials, especially under various harsh environments, for example, in maritime environments, was reported to be obviously lower than conventional concrete [14]. Therefore, developing novel and sustainable cementing materials is still a scientific challenge for the sustainability of marine development.

In addition, cementitious materials are reported by previous research [15–17] to be susceptible to a range of physical and chemical breakdown processes in maritime environments. In addition, the use of standard construction materials (freshwater and river sand, etc.) on most islands and reefs in marine engineering is always constrained by time, transportation costs, and difficult geological conditions [17]. Therefore, how to obtain alternative and sustainable construction materials locally and economically is a significant challenge [18,19].

High-water material (HWM), also known as high-water back-filling material or high-water-content and quick-setting material, is a novel cementing material. HWM was first invented by Professor Henghu SUN from China University of Mining and Technology in 1989 and successfully used in the coal mining back-filling engineering practice [20]. HWM has been praised as green and sustainable cementing material with numerous excellent advantages, including high water content, good pumpability, rapid setting, high early strength, recrystallization recovery strength after the early failure of the stone body, a simple material production process, and low cost [21]. The chemical composition and mineral composition of the raw material of HWM were analyzed by Xie and Liu (2014) [22], and the hydrating and hardening mechanism of HWM was revealed by Xia et al. (2018) [23]. The basic physical and mechanical properties of HWM were tested by a large number of scholars, and the uniaxial and triaxial compressive strength and creep properties were measured as well. Xie et al. (2013) investigated the influence of curing time on the properties of HWM and revealed that the strength of HWM stones increased with the curing time [24]. Zhang et al. (2017) discussed the effects of water-cement ratios on the physical and mechanical characteristics of HWM [25]. Zhou et al. (2017) conducted an experimental study to investigate the failure characteristics of HWM under loading and divided the stress-strain curve of HWM, which had initial deformation, elastic, plastic deformation, and disruption four stages [26]. The research above proved that HWM is a promising cementing material with excellent physical and mechanical properties.

Initially, HWM was utilized mostly for back-filling, roadway support, etc., in underground coal mining [27–30]. Recent studies indicate that HWM has a promising application in maritime environments. HWM is lauded for its ability to "convert water into stone" because of its ultrahigh water-cement ratio and water content, which can exceed 10:1 and 95%, respectively, and its strength can exceed 5 MPa [20], which means less cementing materials are consumed, hence less $CO_2$ emission is produced. Given that water could be acquired locally and HWM slurry could be transported by pumping, HWM construction is practical, quick, and inexpensive in maritime environments [31].

Previous research has examined the behavior of HWM and demonstrated its usefulness in marine environments. Hou et al. (2012) recommended using HWM for the preservation and reinforcement of coral sand islands and reefs, as well as port reinforcement, and they measured the properties of HWM in marine environments [32]. The hydration mechanism, physical and mechanical properties, and port engineering application of HWM in maritime environments were examined [33]. He et al. (2014) discuss the environmental influences on the physical and mechanical properties of HWM cured in seawater [34]. However, there are still a number of scientific obstacles to implementing HWM in marine engineering. The majority of previous studies focused on the characteristics of HWM mixed with freshwater [34–37]. Given the shortage of fresh water in marine environments, HWM derived from marine sources (i.e., seawater) is viewed as an efficient and cost-effective alternative [38,39]. It is required to explore the viability of preparing HWM with seawater and the impact of the seawater environment on HWM's behavior.

In this study, the physical and mechanical characteristics and microstructure, and chemical composition of HWM produced with seawater were studied.

## 2. Materials and Methods

The raw materials of HWM used in this study consist of two parts: main material (A material, B material) and subsidiary material (A-A material, B-B material). Their chemical compositions are shown in Table 1. The main components of A material are bauxite. B material is composed of lime, gypsum, etc. The main components of A-A material are suspension agent, coagulant, and dispersant, including $Na_2CO_3$, $BaBiO_3$, etc. B-B material is composed of early strength agents, suspension dispersants, etc., including $SiO_2$, $CaSO_4$, etc. The proportion of ingredients employed in this study was A:A-A:B:B-B = 1:0.1:1:0.04, which follows the material formulation proposed by Sun and Song (1994) [20].

**Table 1.** Mineral components of high-water material used in this study.

| ID | Materials | Main Compositions |
|----|-----------|-------------------|
| A | Bauxite | $3CaO \cdot 3Al_2O_3 \cdot CaSO_4$, $2CaO \cdot SiO_2$ etc. |
| A-A | Additives | $Na_2CO_3$, $BaBiO_3$, etc. |
| B | Lime, gypsum, etc. | $CaSO_4$, $CaSO_4 \cdot 2H_2O$ $CaSO_4 \cdot 0.5H_2O$ etc. |
| B-B | Additives | $SiO_2$, $CaSO_4$, etc. |

Figure 1 shows the preparation procedures for preparing HWM. It can be seen that A and A-A were mixed together first, and then enough water was added and stirred thoroughly to produce A seriflux. At the same time, B and B-B were mixed together, and then enough water was added and stirred thoroughly to produce B seriflux. A and B seriflux have a strong fluidity and keep good mobility as a liquid for more than 24 h. Therefore, A and B seriflux have excellent pumpability and can be transported by pipeline, which is convenient for construction in engineering practice. Then A and B seriflux were mixed together and stirred thoroughly to produce HWM stone. HWM has the characteristics of quick setting and early strength. HWM could be solidified in half an hour after mixing A and B seriflux together. The early strength of HWM stone could be over 2 MPa in 2 h, which is about 20% of the final strength, and could increase to more than 60–90% of final strength after 7 days, according to Sun and Song (1994) [20]. After that, the strength of HWM would increase flat. HWM stone should be cured in water for more than 28 days to get final strength. In addition, it was reported that the crystals of HWM could keep growing for a long term hence the mechanical strength of HWM could be recovered after the early failure of the stone body of HWM.

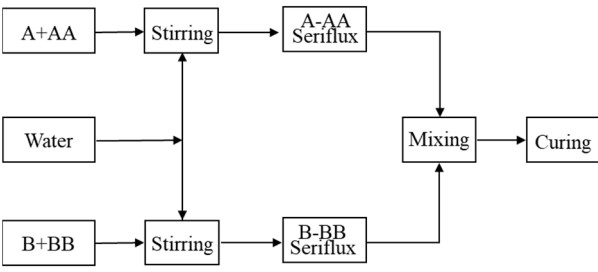

**Figure 1.** Procedures of preparing for HWM.

Table 2 shows the test plan in this study. The HWM samples were prepared and tested according to the Standard for the Test Method of Mechanical Properties of Ordinary Concrete (GB/T 50081-2002) [40] and Methods for Determination of Physical and Mechanical Properties of Coal and Rocks (GB/T 23561.1-2009) [41]. By varying the amount of water added, standard cylindrical HWM samples with a diameter of 50 mm and a height of 100 mm (50 mm × 100 mm) with different water-cement (w/c) ratios (i.e., w/c = 3:1, 4:1, 5:1, and 6:1) were produced. The HWM samples prepared with and cured in tap water were used as the control test group (Test ID: C). As a comparison, artificial seawater was used to prepare and cure samples of the seawater test group (Test ID: S). The artificial seawater

used in this study was manmade in the laboratory by following ASTM D1141-98(2013) [42]. The temperature of the fresh water and seawater was about 18–22 °C.

**Table 2.** Specimen design and test plan.

| Test ID | Test Group | Mixing and Curing Water | w/c | Parameters Investigated |
|---|---|---|---|---|
| C | Control test group | Tap water | 3:1, 4:1, 5:1 and 6:1 | Density, water content, strength, microstructure, chemical component etc. |
| S | Seawater test group | Artificial seawater | | |

Note: S3-2 means the 2nd HWM sample of the seawater test group with w/c = 3:1.

In accordance with the procedure proposed by He et al. (2014) [34], HWM was prepared and cured. The HWM samples were prepared in a standard cast iron mold ($\varphi$50 mm × 100 mm). As shown in Figure 1, the preparation procedures of preparing HWM in this study comprised weighing, mixing with water, combining A and B seriflux, injecting into the mold, and demolding. After demolding, the samples were then placed in tap water or seawater and cured for 28 days. All samples were put in an environment-controlled room at Sichuan University. The room temperature was kept at 26–30 °C, and the air humidity was 40–50%.

The samples were then utilized in the subsequent analysis. After analyzing the microstructure and chemical components with a scanning electron microscope (SEM) and X-rays, the fundamental physical characteristics, such as moisture content and density, were determined. MTS815.03 Electro-hydraulic Servo-controlled Rock Mechanics Testing System was used to test the mechanical properties of samples. Each group sample was subjected to the aforementioned tests three times, with the average value being the final test results. The above experiments were carried out in the State Key Laboratory of Hydraulics and Mountain River Engineering of Sichuan University, China.

## 3. Results

### 3.1. Density and Water Content

The density of HWM produced from water and seawater at various w/c ratios is depicted in Figure 2. It can be seen that as the w/c ratio grows, the average mass and bulk density of samples from two groups of HWM samples drop. As the w/c ratio increases from 3:1 to 6:1, the density of HWM produced from seawater decreases from 1.25 to 1.12 g/cm$^3$. The density steadily drops until it reaches the density of water (1 g/cm$^3$). However, the density of HWM prepared from seawater with the same w/c ratio is 0.8% to 2.2% more than that of HWM prepared from water.

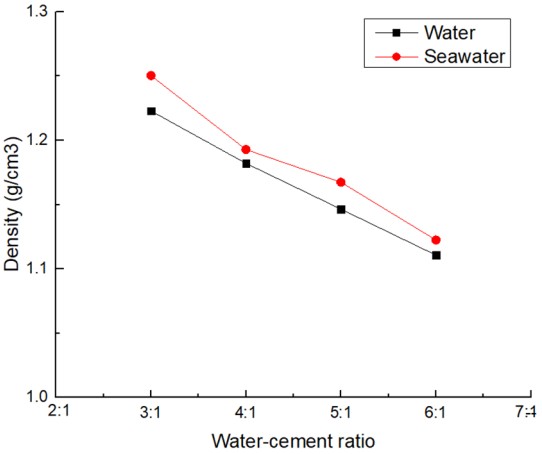

**Figure 2.** Density of HWM with various w/c ratios.

Figure 3 depicts the water content of HWM produced by mixing freshwater and seawater at various w/c ratios. As the w/c ratio increases, the moisture content of the two groups of HWM samples increases progressively. As the w/c ratio increases from 3:1 to 6:1, the water content of HWM produced from seawater increases from 72.1% to 77.5%. The HWM stone body with the same w/c ratios as the seawater test group had 2–10% less water than the control test group. Previous studies also found that when water-cement ratios increase, the density of HWM decreases while the water content of HWM increases, which is consistent with the findings based on the above test results.

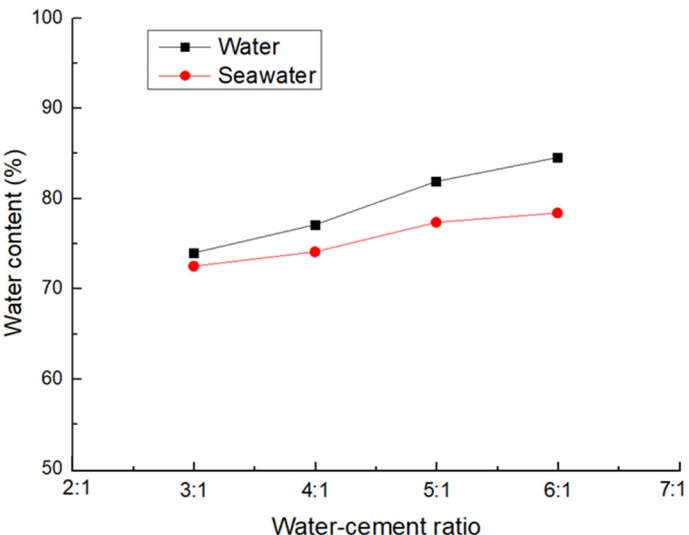

**Figure 3.** Water content of HWM with various w/c ratios.

*3.2. Mechanical Characteristics*

The stress-strain curves of S3 and S6 are shown in Figure 4.

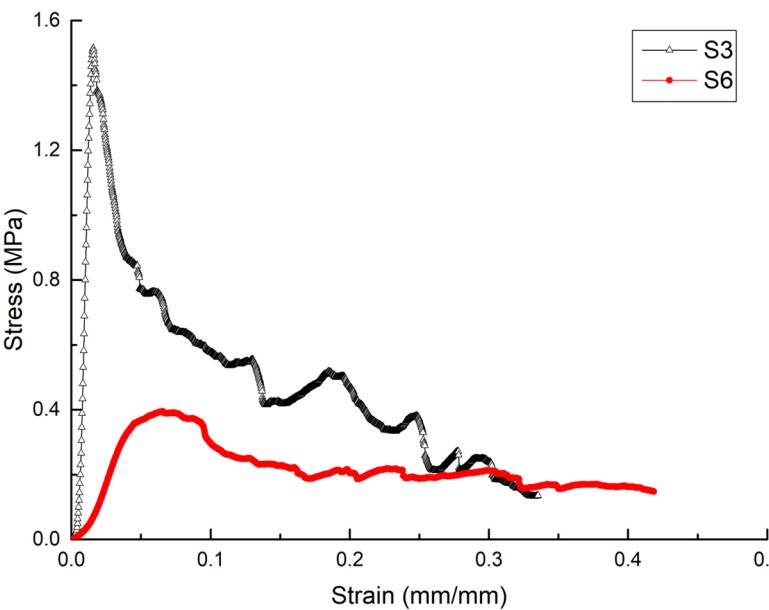

**Figure 4.** Typical strain-stress curves of HWM made by seawater.

In accordance with the rock mechanics classification standard, the stress-strain curve of a typical HWM can be classified into four stages: pore fracture compaction stage, elastic deformation to fracture development stage, unsteady fracture development stage, and post-peak failure stage. Among them, S3 has the shortest pore crack compaction stage, then

enters the elastic deformation stage with a sharply rising stress-strain curve and an elastic modulus of 0.198 GPa. Afterward, S3 quickly reaches its maximum strength (i.e., 1.5 MPa), and the strain is 0.015%. As soon as it enters the post-peak phase, the residual stress is nearly nil.

Sample S6's stress-strain curve demonstrates a substantially longer pore fracture compaction stage and an unstable fracture development stage. In the elastic phase, its stress-strain curve is quite moderate, as the strain increases from 0.02 to 0.04 and the corresponding stress increases from 0.1089 MPa to 0.3276 MPa, while the linear phase elastic modulus is just 0.009 GPa. In the post-peak phase, stress slightly reduces as strain increases. The range of residual stress is 0.08–0.1 MPa, which corresponds to approximately 45–70 percent of the peak stress, while the range of strain is 0.12 to 0.20. It proved that HWM with higher water-cement ratios has better plasticity.

In the uniaxial compression test, Figure 5 depicts the typical failure images of the HWM produced from seawater with w/c = 3:1 (i.e., S3) and 6:1 (i.e., S6). Split failure can be seen to be the failure mode of S3. During loading, first, the stress of S3 sharply spiked while there was no vertical compression observed, after that the sample was split into many pieces in the internal axial direction, after which the stress fell sharply, and the sample could only hold a small axial strain, which is in accordance with the stress-strain curve depicted in Figure 4. At the same time, S6's mode of failure was a ductile failure. Under axial loading, the axial strain continued to grow, and the specimen was compressed vertically and extended laterally. As the axial compression increased, the upper portion of the specimen was initially crushed, and the shattered HWM blocks piled on the upper portion of the specimen and continued to support the load. The sample resembled a "compressed biscuit" in appearance. During this procedure, the material remained somewhat intact, and residual stress remained elevated. Figure 4 depicts the stress-strain curve for the material. A similar phenomenon was also observed in previous studies on the failure mode of HWM [43,44]. It was reported that HWM with higher w/c ratios owns better plasticity; therefore, it could generate a larger axial and horizontal strain under axial loading.

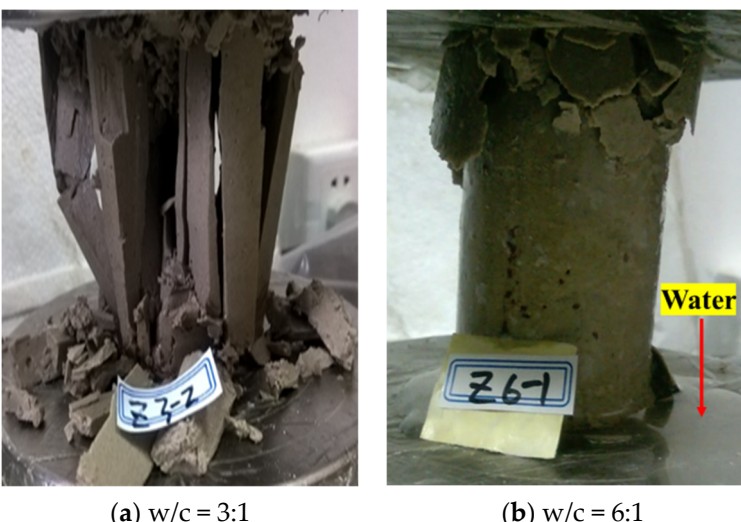

(**a**) w/c = 3:1              (**b**) w/c = 6:1

**Figure 5.** Compression failure mode of HWM made by seawater with various w/c.

During the uniaxial compression test, the surface of the S3 sample remained rather dry, and no water leaked out. While in S6, water precipitated continually, and as the strain increased, water bleeding accelerated. As seen in Figure 5, the bleed water was collected near the sample's base. Sun and Song (1994) proposed that the water in HWM could be divided into three parts, namely crystal water, absorbed water, and free water [20]. Most of the water in HWM is free water, and the amount of free water increases with the

water-cement ratios. Free water is apt to be lost under loading. Therefore, a large amount of water was observed to be leaked out for HWM with higher water-cement ratios.

Two groups of HWM samples were then subjected to uniaxial compression tests to obtain the uniaxial compression strength. The loading rate for the tests was kept at 5 mm/min. Figure 6 depicts the sample strength of two distinct groups. It can be shown that as the w/c increased, the uniaxial compressive strength of the HWM samples in the two groups fell gradually. The sample strengths of the seawater test group with w/c = 3:1 and 6:1 were 1.47 MPa and 0.39 MPa, respectively, while those of the control test group were 1.30 MPa and 0.31 MPa, respectively. The HWM sample strengths of the seawater group were 13 to 60 percent higher than those of the freshwater group of HWM samples, respectively.

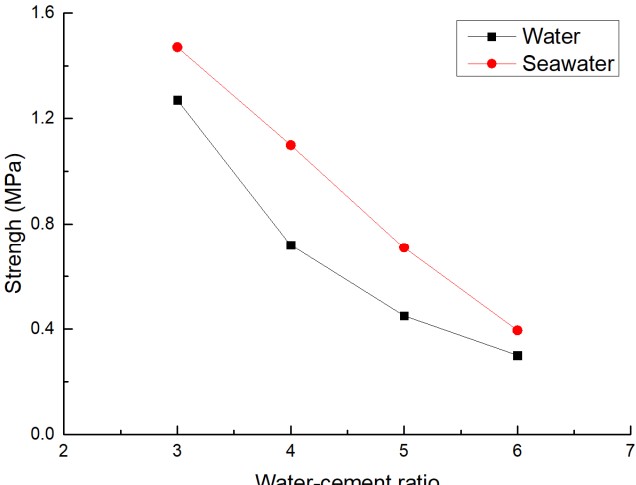

**Figure 6.** Strength of HWM made by seawater and water.

*3.3. Microstructure and Chemical Compositions Analysis*

Figure 7a–c show SEM images (×2000 times) of HWM samples of C3, S3, and S6, respectively.

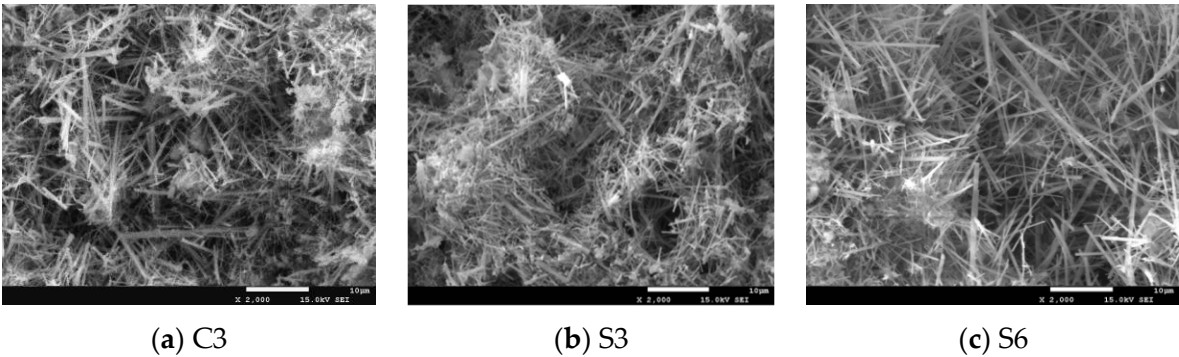

|  |  |  |
|:---:|:---:|:---:|
| (**a**) C3 | (**b**) S3 | (**c**) S6 |

**Figure 7.** SEM photos (×2000) of HWM made by water and seawater.

Figure 7a demonstrates that the hydration products of HWM produced by freshwater are dominated by ettringite crystals and have a needle-like, net-like, and predominantly rod-like structure. The crystals are staggered and interconnected to produce a dense network structure that serves as a framework and support. Filling the network structure with fibrous hydrated silica (C-S-H) gel and pom-shaped aluminum (Al(OH)$_3$) glue reduces the number of interior pores and increases the density. This explains why the water-cement ratio and water content of HWM are so high.

Comparing SEM images of HWM made by water and seawater with w/c = 3:1 (i.e., Figure 7a C3 and Figure 7b S3, respectively), the microstructures of both groups include a network structure composed of ettringite crystals, as well as fibrous hydrated silica (C-S-H) gel and pom-shaped aluminum (Al(OH)$_3$) glue within the network structure. However, there are more ettringite crystals within, and the network structure is significantly denser in HWM generated from seawater, which increases the network structure's strength in the HWM-hardened body. In addition to C-S-H gel and Al(OH)$_3$ glue, a significant amount of M-S-H gel and Mg(OH)$_2$ were found to be present in the network structure, which makes the structure denser. The strength of HWM with a stronger and denser microstructure is always greater. Consequently, the differences in the SEM images of HWM created using fresh water and seawater are consistent with the findings of Figure 6.

By comparing Figure 7b,c, the differences in the SEM images of HWM produced by seawater with a w/c ratio of 3:1 and 6:1 can be determined. It can be observed that HWM with w/c = 6:1 has significantly fewer ettringite crystals, as well as C-S-H gel, Al(OH)$_3$ glue, M-S-H gel, and Mg(OH)$_2$ inside the crystal network structure. HWM with a w/c ratio of 6:1 is able to absorb more water and has a larger water content than HWM with a w/c ratio of 3:1. This is because the network structure is significantly looser and has many more internal pores. In addition, the ettringite crystals of w/c = 6:1 are thinner, and the fibrous hydrated silicic acid gel and pom-shaped aluminum glue in the network structure are also significantly reduced, resulting in a structure with less compactness and more holes.

The C3 and S3 XRD scans are depicted in Figure 8. Ettringite (chemical formula: 3CaO•Al$_2$O$_3$•3CaSO$_4$•32H$_2$O), C-S-H, and Al(OH)$_3$ are the primary constituents of the stone body of the HWM, as seen. In addition to the aforementioned component, however, M-S-H gel and Mg(OH)$_2$ were also detected in the HWM of the seawater group.

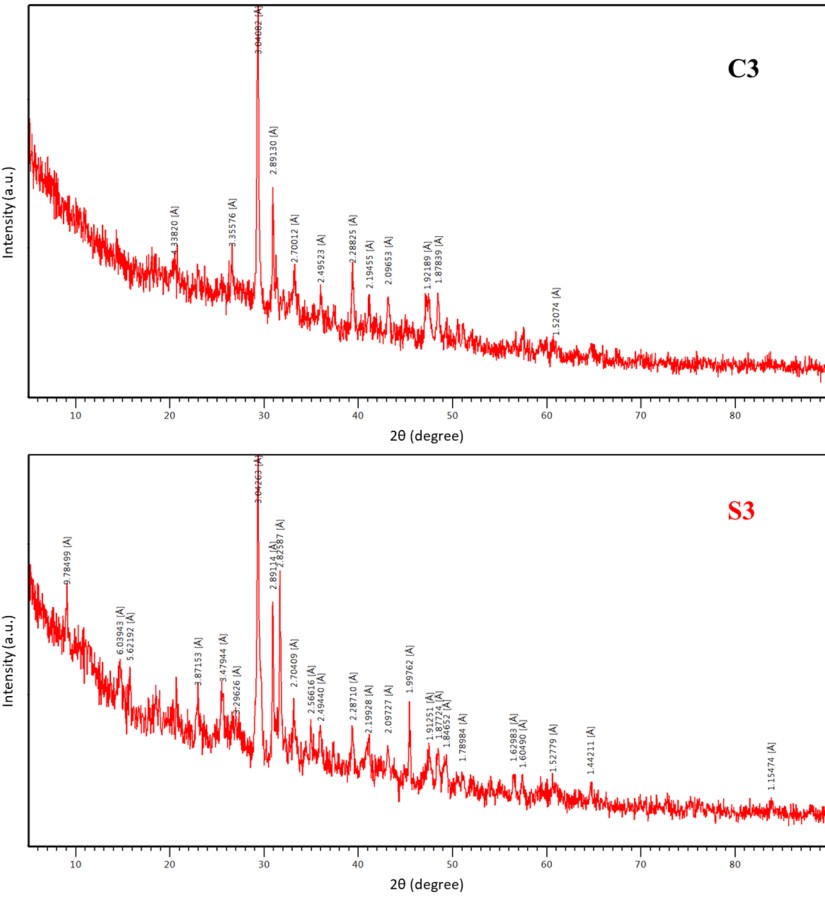

**Figure 8.** XRD image of C3 and S3.

## 4. Discussion

### 4.1. Effects of w/c on the Microscopic and Macroscopic Properties of HWM Made by Seawater

In comparison to fresh water, seawater has a substantial impact on the microstructure, chemical content, and physical and mechanical properties of HWM.

Zhang et al. (2017) analyzed the microstructure of HWM produced using potable water [25]. Figure 7 shows that compared to the SEM pictures of HWM made with fresh water, HWM made with seawater has a denser microstructure, a greater number of ettringite crystals, and a smaller porosity. This is compatible with the impacts of seawater on the macroscopic properties of HWM, specifically a decrease in water content, an increase in density, and an increase in strength.

Comparing the microstructure of HWM with different w/c ratios, the principal hydration products of HWM produced by seawater are the needle-like and prismatic Ettringite crystals, C-S-H and M-S-H gel, $Al(OH)_3$ glue, and $Mg(OH)_2$. Ettringite crystals in HWM with higher w/c ratios are narrower than those with lower w/c ratios.

In addition, when w/c ratios grow, fewer ettringite crystals are produced, much less fibrous hydrated silica gel and pom-shaped aluminum glue are filled in the network structure, and HWM has a looser microstructure and more interior pores. Comparing the macroscopic parameters of HWM with different w/c ratios, as illustrated in Figures 2–4 and 6, HWM formed from seawater with a greater w/c ratio has higher water content, lower density, and lower strength. Therefore, the microscopic and macroscopic properties of HWM in our investigation are compatible.

Additionally, w/c has a substantial impact on the failure modes of HWM. During the uniaxial compression test, hardly any water separated from samples with a w/c ratio of 3:1. The samples demonstrate brittle elasticity, and split failure is the predominant form of failure. The samples with a w/c ratio of 6:1 contain more water (78.4%). During the uniaxial compression test, there was water bleeding, and the rate of water bleeding increased as stress increased. The major mode of failure for samples with a w/c ratio of 6:1 is ductile failure. HWM with a greater w/c has excellent plasticity, whereas HWM with a lower w/c has excellent elasticity-brittleness. This may be the primary reason why failure modes of HWM with different w/c ratios vary.

He et al. (2014) studied the physical and mechanical qualities of HWM manufactured with fresh water but cured in seawater for 21 days. The results demonstrated that HWM cured in seawater could retain outstanding physical and mechanical properties [34]. Hou et al. (2012) conducted field testing with HWM derived from fresh water in maritime engineering, and the HWM demonstrated excellent performance [33]. In this investigation, it was determined that HWM might be prepared and cured in seawater while retaining its outstanding physical and mechanical qualities. HWM is, therefore, a promising building material for islands, reefs, and marine engineering.

### 4.2. Effects of Seawater on the Physical and Mechanical Properties of HWM

Xia et al. (2018) reported that the following reactions will occur after the mixing of two slurries during the hydrating and hardening process of HWM, as shown in Equations (1)–(4) [23]. During the above process, a large amount of $3(3CaO \cdot Al_2O_3 \cdot CaSO_4 \cdot 12H_2O)$ (AFm) and $3CaO \cdot Al_2O_3 \cdot 3CaSO_4 \cdot 32H_2O$ (AFt) were produced.

$$3CaO \cdot Al_2O_3 \cdot CaSO_4 + 2CaSO_4 + 38H_2O \rightarrow 3CaO \cdot Al_2O_3 \cdot 3CaSO_4 \cdot 32H_2O + 4Al(OH)_3 \quad (1)$$

$$3CaO \cdot Al_2O_3 \cdot CaSO_4 + 18H_2O \rightarrow 3CaO \cdot Al_2O_3 \cdot CaSO_4 \cdot 12H_2O + 4Al(OH)_3 \quad (2)$$

$$3Ca(OH)_2 + 3CaSO_4 + 2Al(OH)_3 + 2H_2O \rightarrow 3CaO \cdot Al_2O_3 \cdot CaSO_4 \cdot 32H_2O \quad (3)$$

$$2CaO \cdot SiO_2 + nH_2O \rightarrow C\text{-}S\text{-}H + Ca(OH)_2 \quad (4)$$

It can be observed that a significant amount of the free water in the mixed slurry of A and B dissipates during the hydration reaction and converts into bound water of AFm and AFt, with AFt being the majority of the final reaction result (i.e., ettringite). During the

hydration and hardening reaction of HWM, the formation of Ettringite crystals is largely reliant on the amount of water, with free water forming in HWM stones with a higher w/c ratio and less bound water forming in ettringite with a low w/c ratio. The ideal w/c ratio for HWM made using potable water is 6.86, and the HWM water content is 87.3% [24].

Seawater provides a high concentration of salt, with $SO_4^{2-}$ and $Mg^{2+}$ being the most influential ions on the hydration and hardening reactions of HWM.

According to reports [45,46], $SO_4^{2-}$ reacts with $Ca^{2+}$ to generate gypsum ($CaSO_4 \cdot 2H_2O$). On the one hand, gypsum can create ettringite (AFt) directly with tricalcium aluminate (C3A) in nail slurry. $CaSO_4$ must be present for AFm to react with gypsum dihydrate and form more ettringite (AFt). Equations (5)–(8) depict the primary chemical reaction equations for the procedure described above.

$$Ca(OH)_2 + Na_2SO_4 + 2H_2O \rightarrow CaSO_4 \cdot 2H_2O + 2NaOH \tag{5}$$

$$3(CaSO_4 \cdot 2H_2O) + 3CaO \cdot Al_2O_3 + 26H_2O \rightarrow 3CaO \cdot Al_2O_3 \cdot 3CaSO_4 \cdot 32H_2O \text{ (AFt)} \tag{6}$$

$$3CaO \cdot Al_2O_3 \cdot 3CaSO_4 \cdot 32H_2O + 3(CaSO_4 \cdot 2H_2O) \\ +4H_2O \rightarrow 3(3CaO \cdot Al_2O_3 \cdot CaSO_4 \cdot 12H_2O) \tag{7}$$

$$2(CaSO_4 \cdot 2H_2O) + 3CaO \cdot Al_2O_3 \cdot CaSO_4 \cdot 12H_2O + \\ 16H_2O \rightarrow 3CaO \cdot Al_2O_3 \cdot 3CaSO_4 \cdot 32H_2O \tag{8}$$

As HWM is a novel cement-based material, its hydration and hardening processes are comparable to those of conventional cement-based materials, such as cement and concrete. When concrete is placed in seawater, ion exchange may occur, according to previous studies [47]. $Ca^{2+}$ in C-S-H and $Ca(OH)_2$ would be replaced by $Mg^{2+}$, resulting in the formation of Mg-S-H gel and insoluble $Mg(OH)_2$. Equations (9)–(11) [46,48] depict the principal chemical reaction equations of the aforementioned procedure for the production of HWM from seawater. It can be observed that AFm combines with $MgSO_4$ to form more ettringite AFt, hence increasing the number of ettringite crystals in the body of the high-water material after it has been hardened. M-S-H gel and insoluble $Mg(OH)_2$ were used to strengthen the compactness of the hardened body of HWM by filling the network structure of Ettringite crystals. Therefore, it contributes to the enhancement of the tough body of HWM.

$$Ca(OH)_2 + MgSO_4 + 2H_2O \rightarrow CaSO_4 \cdot 2H_2O + Mg(OH)_2 \tag{9}$$

$$\text{C-S-H} + MgSO_4 + 2H_2O \rightarrow \text{M-S-H} + CaSO_4 \cdot 2H_2O \tag{10}$$

$$4CaO \cdot Al_2O_3 \cdot 13H_2O + 3MgSO_4 + 2Ca(OH)_2 \\ +20H_2O \rightarrow 3CaO \cdot Al_2O_3 \cdot 3CaSO_4 \cdot 32H_2O + 3Mg(OH)_2 \tag{11}$$

In conclusion, $SO_4^{2-}$ and $Mg^{2+}$ in seawater can enhance the synthesis of additional ettringite AFt in the hydration and hardening reaction of HWM, hence enhancing the strength of the hardened skeleton structure. In addition, M-S-H gel and insoluble $Mg(OH)_2$ were produced to fill the internal holes of the ettringite network structure. This improved the density of the hardening body of high-water material, hence enhancing the strength of seawater-based HWM.

## 5. Conclusions

In this study, HWM was first proposed to be produced from seawater and used in a maritime environment. Then the method of HWM produced from seawater was provided, and the physical and mechanical characteristics of HWM made from seawater were investigated, and the interaction mechanism between HWM and seawater was also discussed. The subsequent findings were reached:

(1) As the w/c ratio increases from 3:1 to 6:1, the water content, density, and uniaxial compressive strength of HWM produced from seawater varied from 72.1% to 77.5%; 1.25 to

1.12 g/cm$^3$, and 1.47 MPa to 0.39 MPa, respectively, which is 2–10% lower, 0.8–2.2% higher, and 13–45% stronger than that from fresh water. Compared to HWM samples made with fresh water, seawater test group samples show a 0.8–2.2% greater density, 2–10% lower water content, and 13–45% greater strength.

(2) The primary chemical components of HWM derived from seawater are Ettringite, C-S-H gel, aluminum (Al(OH)$_3$) glue, M-S-H gel, and Mg(OH)$_2$. Ettringite crystals stagger and link, forming a dense network structure that functions as a skeleton and support. The network structure was filled using C-S-H gel, aluminum (Al(OH)$_3$) glue, M-S-H gel, and Mg(OH)$_2$.

(3) The HWM-seawater interaction was revealed. SO$_4^{2-}$ and Mg$^{2+}$ in seawater contribute to the hydrating and hardening reaction of HWM, resulting in the production of additional Ettringite, M-S-H gel, and Mg(OH)$_2$, which makes the skeletal structure of HWM denser, hence enhancing its strength. HWM produced from seawater retains outstanding physical and mechanical characteristics.

(4) This study demonstrated that HWM produced from seawater possesses excellent physical and mechanical properties; consequently, HWM is a promising construction material for islands, reefs, and marine engineering, and it is suggested that additional field tests be conducted to verify the long-term behavior of HWM produced from seawater.

**Author Contributions:** Formal analysis, N.F.; Writing—original draft, B.L.; Writing—review & editing, C.L., J.G. and N.F. All authors have read and agreed to the published version of the manuscript.

**Funding:** This research was funded by the Geological Survey Program of China Geological Survey (Grants No. DD20190269, DD20221782).

**Institutional Review Board Statement:** Not applicable.

**Informed Consent Statement:** Not applicable.

**Data Availability Statement:** Not applicable.

**Conflicts of Interest:** The authors declare no conflict of interest.

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
