# Peer review of "Study on Physical and Mechanical Properties of High-Water Material Made by Seawater"

_sustainability, doi:10.3390/su15043334_

Round 1

Reviewer 1 Report

In this paper, the authors propose the viability of HWM manufactured from seawater and explore its physicomechanical characteristics. In general, this article is well-organized, and all supplied information is pertinent. I believe the manuscript will attract a wide readership of Sustainability. Before publishing, the manuscript still needs some minor adjustments.

1. Please review and comment on the possible benefits of HWM use in islands, reefs, and marine engineering in the Introduction section, so that readers may grasp the value of this study to the field of research. In addition, the Introduction section should be revised to provide references to more current publications in leading journals.

2. In Lines 71-74, I am unsure whether the approach for preparing HWM samples is standard or unique. If this is a standard procedure, then references should be provided.

3. The authors stated in Line 241 that "ions exchange may occur when concrete is added to saltwater". It is difficult for the reader to comprehend the writers' reasoning if the similarity between concrete and HWM is not first explained. According to my knowledge, concrete and HWM are both cement-based materials, and the hydration and hardening processes are comparable.

4. The article has several language errors. For instance, in Line 69, "man-made" is inappropriate. Artificial seawater may be preferable to manmade or manmade seawater. Line 62 should read "the second HWM sample" instead of "the second HWM sample." Line 86 should be updated from "1g/cm3" to "1g/cm3". The authors should revise their entire manuscript for grammatical problems.

Author Response

We thank the reviewer for sparing your precious time to review the manuscript. The reviewer's comments were constructive, and we appreciate such constructive feedback on our manuscript. We have now revised the manuscript carefully based on your concerns. A list of point-to-point responses was prepared, as shown below.

Reviewer 2 Report

Review Assessment for Sustainability-2084711:

The authors conducted an experimental study on physico-mechanical properties of high-water material made by seawater. The paper could be published in the Sustainability, after taking the following comments into consideration:

1)     There are some grammatical and punctuation errors that must be addressed. A comprehensive English proofread is needed.

2)     The novelty of this study must be clarified in the abstract.

3)     Abstract: Please summarize the main points and avoid unnecessary parts. Add some of the most critical quantitative results to the Abstract.

4)     Introduction should be like a funnel and it is required that you start from general to specific points. So, I recommend that the literature review can be improved. In order to give a better overview of the importance of research on building materials issues and in particular cement and concrete, it is suggested to use the results of other researchers to improve your literature review. For example, some suggested references are: “Cement Paste Modified by Nano-Montmorillonite and Carbon Nanotubes” and "Strength optimization of cementitious composites reinforced by carbon nanotubes and Titania nanoparticles"

5)     The quality of the figures (e.g. Fig. 1, 2, … ) should be improved, and the information in the figures should be more legible.

Author Response

We thank the reviewer for sparing your precious time to review the manuscript. The reviewer's comments are valuable and constructive, and we appreciate such constructive feedback on our manuscript. A list of point-to-point responses was prepared, as shown below.

Reviewer 3 Report

The article "Study on Physico-mechanical Properties of High-water Material Made by Seawater" does not meet the level of Sustainability journal. The paper is of poor scientific novelty, with a large number of inaccuracies and mistakes. The number of methods for evaluating the properties of the resulting HWM is insufficient, the quality of some images is unacceptable (Figure 7), and there are references in the text to non-existent data (like table 4 on line 177). The possible chemical interaction is not proven with experimental data and relies solely on other authors' works.

Author Response

We thank the reviewer for sparing your precious time to review the manuscript. The reviewer's comments are valuable and constructive, and we appreciate such constructive feedback on our manuscript.

Reviewer 4 Report

It seems that the paper was rejected and resubmitted.

But as a reviewer, reading this paper is too difficult.

The authors should also upload unmarked version.

Novelty statement is not enough.

Material method is not clear more detail should be added.

Sustanability for cement and concrete production is an important subject. The authors should add a paragraph for this. The following paper should be added for this purpose: influence of replacing cement with waste glass on mechanical properties of concrete; use of recycled coal bottom ash in reinforced concrete beams as replacement for aggregate; concrete containing waste glass as an environmentally friendly aggregate: a review on fresh and mechanical characteristics; mechanical behavior of crushed waste glass as replacement of aggregates;flexural behavior of reinforced concrete beams using waste marble powder towards application of sustainable concrete; improvement in bending performance of reinforced concrete beams produced with waste lathe scraps; performance assessment of fiber-reinforced concrete produced with waste lathe fibers; performance evaluation of fiber-reinforced concretes produced with steel fibers extracted from waste tire; performance evaluation of fiber-reinforced concretes produced with steel fibers extracted from waste tire 

Processes of material production should be included.

More details should be provided for Fig 4.

More conclusions should be included

Author Response

We thank the reviewer for sparing your precious time to review the manuscript. The reviewer's comments were constructive, and we appreciate such constructive feedback on our manuscript. We have now revised the manuscript carefully based on your concerns. All the corrections we made in the revised manuscript were marked in red font. A list of point-to-point responses was prepared, as shown below. In addition, unmarked version also has been uploaded to the online submission system.

Round 2

Reviewer 2 Report

Review assignment for Sustainability-2084711:

I am satisfied with the changes in the revised manuscript. I recommend the paper to be published after addressing the comment:

Reference #4: the journal name should be modified to "ACI Materials Journal".

Author Response

We thank the reviewer for sparing your precious time to review the manuscript. The reviewer's comments are valuable and constructive, and we appreciate such constructive feedback on our manuscript.

Q1. Reference #4: the journal name should be modified to "ACI Materials Journal".

Response: Thank you very much for your reminder. We have made the corrections suggested by you.

Reviewer 3 Report

Unfortunately it is not possible to assess the changes made by the authors of the work because the manuscript is not properly framed.

Author Response

We thank the reviewer for sparing your precious time to review the manuscript. The reviewer's comments were constructive, and we appreciate such constructive feedback on our manuscript. We have now revised the manuscript carefully based on all reviewers’ suggestions and all the corrections were marked with a red fond.

Q1. Unfortunately it is not possible to assess the changes made by the authors of the work because the manuscript is not properly framed.

Response: Thank you very much for your comments.

The novelty of this work consists of the following aspects:

  1. It was initially proposed that HWM might be made from saltwater and utilized in maritime environments, which gives a way for producing HWM from marine sources.
  2. This work presents laboratory evidence to demonstrate the viability of HWM created from seawater and demonstrates that HWM is a suitable building material for islands, reefs, and marine engineering.
  3. This study reveals the interaction mechanism between HWM and seawater.

We acknowledge that verifying the viability of HWM generated from seawater from all angles is a significant undertaking, and additional research, such as the long-term behavior of HWM in marine environments, should be done in the future. It is difficult for us to address all issues in a single document.

In order to meet the requirements of the Sustainability journal, we meticulously examined the manuscript and made the corrections suggested by all reviewers. Each correction was indicated in red type.

We hope that the reviewer could reconsider our revised manuscript.

Reviewer 4 Report

The paper can be accepted in this form.

Round 3

Reviewer 3 Report

The authors considered most of the comments or adequately responded to the remarks contained in the review; therefore, the work may be approved for publication after editorial and English corrections.